# Multiple Sessions of Antimicrobial Photodynamic Therapy Improve Periodontal Outcomes in Patients with Down Syndrome: A 12-Month Randomized Clinical Trial

**DOI:** 10.3390/dj13010033

**Published:** 2025-01-15

**Authors:** Rafael Ferreira, Sebastião Luiz Aguiar Greghi, Adriana Campos Passanezi Sant’Ana, Mariana Schutzer Ragghianti Zangrando, Carla Andreotti Damante

**Affiliations:** 1Discipline of Periodontics, Faculdade de Odontologia, Universidade Federal de Mato Grosso do Sul, Campo Grande 79070-900, Brazil; rafael_ferreira@ufms.br; 2Discipline of Periodontics, Faculdade de Odontologia de Bauru, Universidade de São Paulo, São Paulo 05508-220, Brazil; slagregh@fob.usp.br (S.L.A.G.); acpsantana@usp.br (A.C.P.S.); mariana@fob.usp.br (M.S.R.Z.)

**Keywords:** Down syndrome, photodynamic therapy, periodontitis, laser

## Abstract

**Background/Objectives:** Individuals with Down syndrome (DS) often present with severe periodontal disease at a young age. Adjuvant treatments to scaling and root planing (SRP), such as antimicrobial photodynamic therapy (aPDT), may benefit this population. This study evaluated the effectiveness of aPDT as an adjunct to SRP in individuals with DS. A randomized, double-blind, parallel trial was conducted with 37 individuals with DS. **Methods:** The test group (aPDT; *n* = 18) received SRP + aPDT, while the control group (C group; *n* = 19) received SRP only. For aPDT, a red laser (658 nm; 0.1 W; 2229 J/cm^2^; 40 s sweeping with optical fiber) combined with methylene blue (MB) (100 µg/mL) was applied across repeated sessions (on days 3, 7, and 14). Clinical parameters, such as plaque index (PI), clinical attachment level (CAL), probing depth (PD), and bleeding on probing (BOP), were recorded at baseline and after 3, 6, and 12 months of treatment. Statistical analyses were performed using parametric and non-parametric tests (*p* < 0.05). **Results:** Both treatments promoted improvements in all clinical periodontal parameters (*p* < 0.05). The aPDT group showed a statistically significant reduction in CAL at 3 months (aPDT = 4.58 mm vs. C = 4.72 mm; *p* < 0.05) and 12 months (aPDT = 4.59 mm vs. C = 4.84 mm; *p* < 0.05). **Conclusions:** aPDT improved periodontal health in the long term through a stable gain in attachment.

## 1. Introduction

Down syndrome (DS) is the most common autosomal chromosomal abnormality, associated with the presence of an extra chromosome 21 (simple trisomy—94.6% of cases), mosaicism (3.1%), and translocation (2.3%) [1]. The prevalence of DS varies between countries, with a ratio of 1:700 to 1:1000 live births [2]. These patients have a high prevalence of periodontal disease (PD) [3,4,5], even when compared to healthy controls or patients with other special needs [6,7]. This high susceptibility to PD cannot be attributed solely to poor or lack of oral hygiene but is also associated with the congenital condition itself, involving genetic mechanisms [8] and immune responses [9,10,11,12,13]. Due to its peculiar characteristics, DS is included in the classification of periodontal disease, defined by the World Consensus as periodontal disease associated with genetic alterations [14]. In the past, it was presumed that individuals with DS did not present a different periodontal microbiota compared to individuals without the syndrome [15]. However, recent studies demonstrate a greater presence of certain periodontal pathogens in these patients [16,17,18,19]. Moreover, even after conventional periodontal treatment, these patients continue to have high levels of bacteria from the red complex [20]. Therefore, the use of adjunctive therapies to scaling and root planing (SRP) is indicated to achieve additional effects. Some available treatments include antimicrobial agents (e.g., chlorhexidine) for continuous use [21,22,23,24], surgical interventions [25], or antimicrobial photodynamic therapy (aPDT) [26]. Additionally, specific and individualized guidelines for oral hygiene should be provided to these patients and their caregivers [27]. However, the use of antimicrobial agents (e.g., chips, fibers, antibiotic mouthwashes, or tablets) also requires patient cooperation with hygiene, as these agents may have varying long-term effects [28]. Antimicrobial photodynamic therapy (aPDT), mediated by the photosensitizer methylene blue, has demonstrated positive results as an adjunct to basic treatment in reducing periodontal pathogenic bacteria [29,30], bleeding on probing [29,31,32,33,34,35,36], probing depth [32,33,35,36,37,38,39], and inflammatory mediators [29,30,32,34,36,40]. However, there is a paucity of studies evaluating the effect of aPDT in patients with DS, and to date, no clinical studies with long-term follow-up have been conducted. A systematic review and meta-analysis did not show statistically significant results in favor of aPDT [41,42]. Although modest statistical gains were observed with the combination of aPDT and SRP, the authors suggested that these data might not represent clinically relevant outcomes [43,44]. Other systematic reviews have demonstrated that aPDT produces positive results when used as an adjunct to SRP in the treatment of chronic periodontitis [43,45] and aggressive periodontitis [46]. Furthermore, aPDT was found to be more effective than systemic antibiotics [47] and provided additional clinical improvements in the treatment of residual periodontal pockets [48]. Despite reports of positive effects of this therapy in patients who are immunocompetent, its use in treating periodontal disease in patients with Down syndrome has not shown superior results compared to conventional treatment [26,42,49]. Considering that patients with Down syndrome are more susceptible to PD and that aPDT has shown promising results as an adjunct in periodontal treatment, this study aims to evaluate the effectiveness of multiple sessions of aPDT as an adjunctive treatment in non-surgical periodontal therapy for patients with DS.

## 2. Materials and Methods

This study was conducted in accordance with the CONSORT guidelines and was registered at ClinicalTrials.gov (NCT02938988—23 October 2013). The study protocol was approved by the Ethics Committee of the Bauru School of Dentistry, University of São Paulo (14045513.5.0000.5417—protocol 386.460). Informed consent was obtained from all legal guardians. Complete medical and dental histories were collected from all participants during the initial phase, based on information provided by their parents and/or legal guardians.

### 2.1. Sample Size Calculation

The sample size was calculated based on probing depth (PD) as the primary outcome measure. Using the mean standard deviation (0.96 mm) from the first 14 patients examined, the sample size was determined to provide 80% power and to detect a significant difference of 1 mm between the groups with a 95% confidence interval (alpha = 0.05). The total sample size was 32 individuals, with 16 in each group.

### 2.2. Inclusion and Exclusion Criteria

The inclusion criteria were a diagnosis of trisomy of chromosome 21 due to non-disjunction (“free trisomy”), age 15 years or older, and the presence of at least one permanent tooth in each quadrant of the mouth exhibiting periodontitis [50,51].

Patients with systemic conditions that could potentially affect periodontal health were excluded. These conditions included a diagnosis of Down syndrome (DS) due to translocation or mosaicism, antibiotic use within 6 months prior to the appointment, long-term use of anti-inflammatory medications (e.g., corticosteroids), non-collaborative patients, other neurological disorders (e.g., autism spectrum disorders and cerebral palsy), uncontrolled diabetes mellitus, hyperthyroidism, hypothyroidism, or menopause.

### 2.3. Clinical Parameters

At baseline and at 3, 6, and 12 months post-treatment, a complete periodontal examination was performed by a single calibrated examiner (RF) using a North Carolina-marked manual periodontal probe (Hu-Friedy, Chicago, IL, USA). The third molars were excluded, and the periodontal clinical parameters included the following: clinical attachment level (CAL), gingival recession (GR), gingival hyperplasia (GH), probing depth (PD), bleeding on probing (BOP) [52], and plaque index (PI) [52].

The calibration of the examiner was performed with a non-syndromic patient presenting periodontal disease and was reassessed within 48 h to calculate the kappa value. For the variables PD and BOP, the lowest kappa value obtained was 0.93, with a percentage of agreement of 95.68%.

Data from teeth extracted during the study were not included.

### 2.4. Blinding

This is a double-blind study. Both patients and the statistician were blinded. The patients lacked the cognitive ability to differentiate between the treatments received. Parents and caregivers were informed about the treatment provided at the end of the study [53]. The researcher who applied the interventions and performed the clinical measurements was the same, ensuring consistency in patient care.

### 2.5. Randomization

After the initial examination, randomization using computer-generated random numbers was applied to classify the individuals into one of the treatment modalities: scaling and root planing (SRP) (C group; *n* = 19) and SRP associated with aPDT (aPDT group; *n* = 18).

To obtain homogeneous groups, patients were stratified according to the severity of periodontal disease [50,51].

### 2.6. Study Design

This randomized clinical trial was a controlled parallel study conducted from October 2013 to March 2018.

All patients received scaling and root planing (SRP) using manual instruments (Gracey curettes; Hu-Friedy, Leimen, Germany). Patients were divided into two groups of 18 and 19 individuals. One group received basic periodontal treatment consisting of SRP (C group). The other group received the same basic treatment, complemented by antimicrobial photodynamic therapy (the aPDT group).

For the aPDT group, a red laser (InGaAlP laser, Therapy XT-DMC—São Carlos, SP, Brazil) was applied in combination with methylene blue dye diluted in deionized water (100 μg/mL) (Sigma-Aldrich, São Paulo, Brazil). The entire procedure was performed without local anesthesia. The pre-irradiation time was 3 min [32,35,54]. The laser was applied inside the periodontal pocket using an optical fiber. The laser parameters were as follows: red laser, 658 nm; area of the conductive tip (0.000314 cm^2^); application at specific sites (mesial, center, and distal) followed by sweeping mode irradiation (medial-distal movements) into the periodontal sulcus or pocket (40 s for buccal and 40 s for lingual sites); total time: 80 s per tooth; adjusted power of the device using the optical fiber (70 mW); energy density (2229 J/cm^2^); total energy per tooth (8 J); power density: 222.9 W/cm^2^.

Figure 1 shows the study design timeline. “Day zero” was when the patient had completed dental care (e.g., dental caries, endodontics) and showed no supragingival calculus. On day zero, in the aPDT group, aPDT was performed on all teeth and repeated after 3, 7, and 14 days [55]. The patients in the control group received SRP throughout the entire mouth on the same dates as the aPDT group. After the last aPDT session (or SRP for the control group), the follow-up period began, with controls at 3, 6, and 12 months (or 104, 194, and 374 days, respectively).

Patients in both groups received supragingival dental prophylaxis every three weeks throughout the study period.

All patients, parents, and/or caregivers participated in motivational sessions and received oral hygiene instructions to ensure they could maintain an appropriate level of oral hygiene. During these sessions, a presentation was given to parents and/or caregivers, correlating specific characteristics of patients with Down syndrome (DS) and periodontal disease. Patients were enrolled in a hygiene program tailored to their individual needs and received reinforcement of these instructions at each appointment. Reinforcement through imitation, pictures, drawings, and presentations was used according to their individual needs.

A rigorous control of supragingival biofilm was established from the beginning of the study. Patients received complete instructions on oral hygiene and SRP. Parents and/or caregivers, in addition to the patients, received reinforcement of oral hygiene measures at each study visit. The brushing technique for patients with DS was based on the Fones method. Supragingival biofilm control was performed every 3 weeks. Reinforcement of oral hygiene was provided according to the individual needs of each patient.

### 2.7. Statistical Analysis

All results were tabulated in Excel 2010 (Microsoft^®^ Corporation, Redmond, WA, USA) and analyzed using Statistica 7.0 for Windows (StatSoft Inc., Tulsa, OK, USA), adopting a 5% significance level.

The Shapiro–Wilk test was used to test the hypothesis of normal distribution for all periodontal variables (PD, BOP, CAL, and PI). In the case of a normal distribution, a Student’s *t*-test was performed for comparison between the two groups. Two-way ANOVA and repeated measures ANOVA were applied for intra-group assessment. In the case of non-parametric data, the Mann–Whitney test was used, or the chi-square test was complemented by the Friedman and Wilcoxon tests for evaluation between the groups and the Kruskal–Wallis test complemented by the Tukey test for intra-group analysis.

## 3. Results

During the study period, 96 patients with Down syndrome (DS) were considered potentially eligible. However, 52 patients were excluded for not meeting the inclusion criteria or for meeting one or more exclusion criteria. Of the forty-four remaining patients, seven did not attend the follow-up appointment. Finally, 37 patients were allocated to the control group (C) or the test group (aPDT) (Figure 2).

Demographic data are presented in Table 1.

Based on the mean values of the periodontal parameters, a significant reduction (*p* < 0.05) in CAL was observed between the groups at 3 and 12 months (Table 2). Differences in CAL and similarities in PD measurements between the groups suggest that aPDT promoted a reduction in gingival enlargement induced by dental biofilm (previously referred to as gingival hyperplasia), primarily with promoting benefits in sites with lower (1–3 mm) probing depths (Table 3).

For better visualization of the periodontal changes according to the evaluation periods, the number of sites was quantified based on intervals of probing depth measurements (Table 3).

There was a statistically significant difference (*p* < 0.05) between the groups at 3, 6, and 12 months for the probing depth range of 1 to 3 mm. These data reveal that a greater number of moderate and deep sites in the aPDT group changed to healthy sites.

## 4. Discussion

Periodontal disease is a condition with high prevalence and severity in patients with Down syndrome (DS) [3,4,5], as confirmed in the present study. Patients had a probing depth (PD) of 4.01 ± 0.96 mm and 4.16 ± 0.93 mm in the aPDT and control (C) groups, respectively. Considering the altered immune system [9,10,11,12,13] associated with deficiencies in oral hygiene habits, adjuncts to conventional periodontal treatment [21,22,23,24,25], such as antimicrobial photodynamic therapy (aPDT) [26,49], are necessary to achieve clinical periodontal improvement. The clinical results of our study revealed, after a 12-month follow-up, periodontal health improvements for both groups, with a greater reduction in clinical attachment level (CAL) in the aPDT group.

Nonsurgical periodontal treatment can improve clinical parameters in patients with DS [20,21,22,23,24,26,49], similar to the findings in our study (*p* < 0.05), with additional benefits in the CAL indices at 3 and 12 months due to the adjunctive use of aPDT. The aPDT protocol was applied to the entire mouth without local anesthesia. This methodological modification was made to treat deep periodontal sites and control both supragingival and subgingival biofilm in healthy areas.

The adjunctive use of aPDT in nonsurgical periodontal treatment for patients without DS has demonstrated an effect on CAL [33,35,38,39,55,56,57], with results maintained up to 12 months [52]. These findings are similar to ours. However, all these studies presented a statistical correlation between decreased PD and CAL values, also showing differences between groups concerning PD. Despite this, our study did not show such a relationship. The slight increase (not statistically significant) in PD at 6 months was probably insufficient to create a significant difference between the groups concerning CAL at this time point. In our study, there were no differences in PD values between the groups at any of the follow-up periods (*p* > 0.05), which contrasts with other studies [32,33,35,36,37,38,39,56,57]. This suggests that the statistically significant differences in CAL are related to changes in clinical measures, resulting in a reduction in gingival enlargement caused by dental biofilm. Some reviews with meta-analysis suggest a significant short-term effect of aPDT on the reduction of CAL [58,59], which is consistent with our results showing improvements in periodontal parameters within 3 months.

When stratifying the number of sites based on the PD range, we observed a decreasing trend in periodontal pocket depths and an increasing trend in the number of sites with PD between 1 and 3 mm. Specifically, at this PD range, there were no statistical differences between the groups at 3, 6, and 12 months (*p* > 0.05), with aPDT showing superiority. This could be related to changes in subgingival biofilm due to modifications in supragingival biofilm, as well as possible effects of laser photobiostimulation [60,61] on gingival tissue. Future studies could benefit from qualitative analyses of the supragingival and subgingival biofilm to properly assess the modifications induced by aPDT, which may influence periodontal clinical parameters.

One of the distinguishing features of our study was the stratification of the number of sites with different probing depth intervals (Table 3). These analyses provide insights into which alterations were more pronounced across different sites and patient disease profiles. The reduction (*p* < 0.05) between groups in the intervals with shallow pockets is an unprecedented result, suggesting that the adjunctive use of aPDT could benefit patients with gingivitis (and reduced periodontium). However, considering the periodontitis profile, clinical changes related to CAL should be interpreted with caution, as considering the entire mouth average might suggest that these data do not represent clinically relevant outcomes.

Despite significant changes in sites with shallow probing depth, no statistically significant differences were found regarding BOP. Our results are consistent with the literature [26,49].

It is important to note that, even after conventional periodontal treatment, bacterial levels from the red complex typically remain elevated in diseased sites in patients with DS [20]. Extrapolating these findings to our study, the presence of residual bacteria may hinder proper periodontal healing, especially since these patients exhibit a deficient immune response [10,11,12,13]. Additionally, supragingival biofilm presence can contribute to reinfection of treated sites [20]. In our study, when stratifying by PD sites, no clinical changes in CAL were observed, especially in areas with increased PD, even among those who received aPDT as adjunctive treatment. Another factor is that basic periodontal treatment is performed over multiple sessions, ideally without local anesthesia, and this is also the case during aPDT application. We observed that the introduction of the optical fiber into the periodontal pocket causes mild discomfort, which, depending on the patient’s cognitive threshold, could compromise the success of the procedure. For example, pronounced discomfort was present in deeper pockets (greater than 5 mm). In these cases, the results may influence the therapeutic progression.

This could result in suboptimal treatment of these areas, as the extent of optical fiber penetration and photosensitizer irrigation was dependent on the patient’s behavioral response. Therefore, even with the use of aPDT, nonsurgical periodontal therapy in patients with DS should be optimized, particularly in areas with residual pockets and increased probing depth, before transitioning to periodontal maintenance therapy. Future studies could benefit from the application via alternative modes of action (such as the transgingival mode) to activate the dye, which may result in less discomfort for the patient, as well as the use of local anesthesia for therapy application in deeper pockets (greater than 5 mm).

Multiple sessions of antimicrobial photodynamic therapy (aPDT) promote better clinical outcomes [31,33,34,36,55,62], reducing clinical parameters up to 6 months after treatment [55]. However, data from a systematic review [63] suggest that repeated applications of aPDT for non-surgical periodontal treatment of residual pockets showed no additional clinical effects on periodontal maintenance, a finding corroborated by our study. Since there is no standardized protocol for laser application parameters, as well as the type and concentration of photosensitizer, comparing studies is challenging, and the results should be interpreted with caution. Furthermore, decontamination at different stages of the oral cavity may represent a source of errors, as recontamination is highly likely between sessions, though it may be lower when compared to a single application. Therefore, future studies could benefit from the application of aPDT at different time points, as outlined in the full-mouth disinfection protocol with chlorhexidine [64], as well as during follow-up visits, such as during periodontal maintenance.

The control of dental biofilm and adherence to treatment in patients with Down syndrome (DS) presents a clinical challenge. Strict periodontal maintenance and prophylaxis every 15 days are factors that influence the results of periodontal treatment, as seen in patients with an incisive/molar pattern of aggressive periodontitis [62], which also applies to patients with DS [27]. Studies have shown that more frequent interventions in patients with DS yield better results compared to specific treatment regimens [23,65], especially when combined with antimicrobial agents such as chlorhexidine [21]. In our study, follow-up appointments were suggested every 21 days; however, not all parents or caregivers adhered to this schedule, which may have compromised the maintenance of clinical results.

Among the limitations of our study, the behavioral and motor challenges typical of patients with DS must be considered, as these may affect both the maintenance of periodontal clinical results and the selection of a blinded evaluator. The difficulty in fine motor coordination and the cognitive level of patients with DS generally hinder their ability to achieve and perceive adequate dental hygiene. It is important to note that guidance was provided to both the patients and their parents/caregivers at all appointments. Hygiene instructions were applied uniformly to all patients. However, the individual needs of each patient were identified and assessed during each follow-up visit. For example, some patients had greater biofilm accumulation in the interproximal areas, and in these cases, the use of dental floss was emphasized. However, socio-cultural factors may have influenced the low adherence to hygiene practices by the parents. At all evaluation periods, there were high values for probing depth (PD), clinical attachment level (CAL), plaque index (PI), and bleeding on probing (BOP), complicating the classification of the clinical picture of periodontal health [14]. Since the participants were young patients or young adults, the supervision and support from their parents or caregivers were insufficient. Consequently, oscillations in the periodontal indices were observed in both groups during the different evaluation periods (3, 6, and 12 months). For the BOP, PI, and CAL indices at 6 months, both groups showed an increase (with no statistically significant difference between the groups), consistent with clinical worsening when compared to the 3-month period. This may have impacted the improvement of CAL during this period. These fluctuations highlight the difficulty in maintaining the results achieved during the active treatment phase and emphasize the importance of biofilm control during the periodontal maintenance phase, particularly at home.

The higher PI observed at all time points reflects the typically compromised periodontal status of these patients. Our data are also in agreement with another study [64] that showed an increase and worsening of the PI after periodontal treatment in patients with DS.

Therefore, additional strategies for biofilm control and reinforcement of instructions—such as the use of antimicrobial mouth rinses, application of disclosing agents at home, and greater collaboration from parents and caregivers—should be addressed in future studies.

A limitation of the present study was the lack of a blinded evaluator. This was due to the management strategy, as patients with DS did not accept having a different professional conduct the evaluations. Nonetheless, this cognitive deficit in patients does not affect the blinding of the patient, as they are unlikely to distinguish between the types of treatment. This may generate a beneficial motivational effect, as the patients may perceive the treatment as the same, regardless of the therapy type used. During the pilot phase (the data and participants were not included in this study), there was significant participant refusal to have another operator, which would have made our study triple-blind (participants/assessor/statistician blinded to the procedures). Although our sample was predominantly composed of adults, the cognitive level of these participants required continuous behavioral management maneuvers. Having a researcher present solely for clinical measurements could have conditioned participants to associate a person with the sensation of discomfort. In our study, the same researcher performed both the interventions and the clinical measurements. This condition was necessary due to the clinical management of the participants. Therefore, to mitigate this issue, we opted for maintaining a single operator. However, it is suggested that future studies attempt to apply a triple-blind model, providing a more objective assessment of clinical and therapeutic outcomes.

The application of aPDT to the full mouth via optical fibers inserted into the periodontal sulcus or pockets caused mild discomfort, which required multiple appointments to complete the treatment at all sites/teeth. Despite this, patients with DS often showed good acceptance of this therapy [26,49]. However, physical sessions (such as the use of mouth openers, laser protection glasses, and demonstration of dye irrigation with a syringe) and psychological conditioning (behavioral management regarding the duration of therapy) are required for behavioral adequacy and the subsequent applicability of the technique. Furthermore, anatomical characteristics (such as pseudo-macroglossia or accumulation of saliva in the oral cavity) hinder the optimization of aPDT sessions. These sessions must also take into account the patient’s collaboration and attention span (which was approximately 30 min), with the full application protocol used in our study, lasting on average 45 min.

For future randomized clinical studies, measures should be implemented to reduce bias introduced by cognitive and motor deficits, which impact the oral hygiene of these patients. The use of electric toothbrushes for patients with DS and three-headed manual toothbrushes for caregivers should be encouraged. Patients can perform dental hygiene [60] but should always be supervised and assisted by parents or caregivers, regardless of the patient’s age [27]. The goal is to reduce periodontal pathogens and improve clinical periodontal parameters, particularly during the home periodontal maintenance phase. Additionally, future studies should aim to include larger sample sizes and explore the continuous or extended use of antimicrobial solutions (such as chlorhexidine) in various forms (e.g., sprays, mouthwashes, or slow-release local devices). Reduced intervals between follow-up appointments and the inclusion of multiple aPDT sessions during periodontal maintenance visits should also be considered.

## 5. Conclusions

Both treatments resulted in significant clinical improvements in the non-surgical management of periodontal disease in patients with DS. aPDT contributed to improved periodontal health, with a stable gain in attachment lasting up to 3 and 12 months and promoted benefits in the number of sites that changed to healthy (i.e., the number of sites with shallow probing depths).

## Figures and Tables

**Figure 1 dentistry-13-00033-f001:**
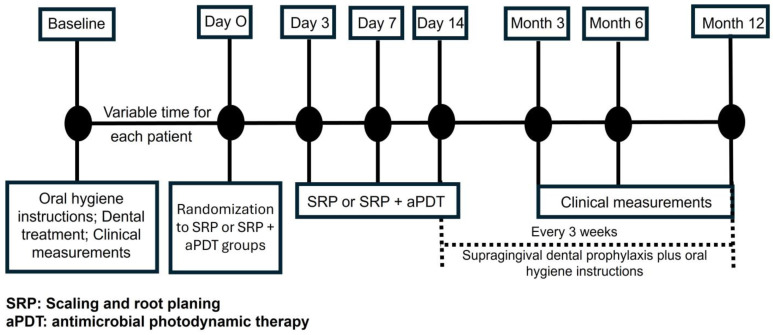
Study design timeline.

**Figure 2 dentistry-13-00033-f002:**
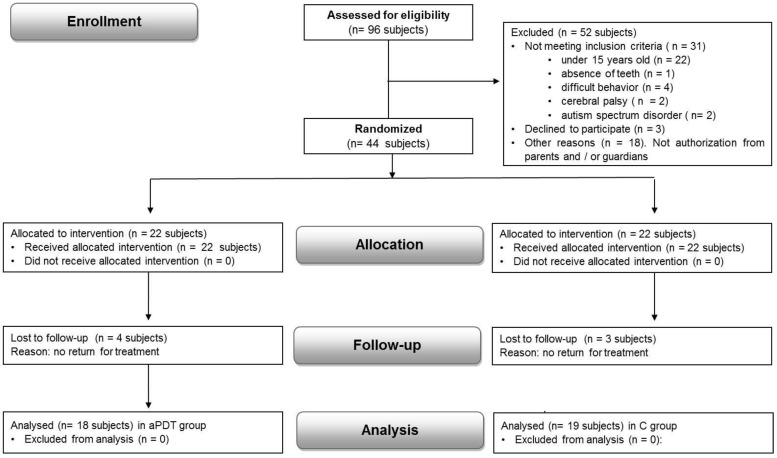
CONSORT flow diagram of the study showing randomization, allocation, and interventions. C—control group: scaling and root planing; aPDT—scaling and root planing + antimicrobial photodynamic therapy.

**Table 1 dentistry-13-00033-t001:** Sample distribution by age, sex, and number of evaluated teeth.

Group	Mean Age (Years)	Sex		Ethnicity		Number of Evaluated Teeth (n)
		Male (%)	Female (%)	White (%)	Black (%)	
aPDT	29.44 ± 7.82	7 (38.9%)	11 (61.1%)	11 (61.1%)	7 (78.9%)	23.89 ± 5.75
C	27.21 ± 6.92	8 (42.1%)	11 (57.9%)	13 (68.4%)	6 (31.6%)	23.11 ± 6.17
Total	28.42 ± 7.41	15 (40.5%)	22 (59.5%)	24 (65%)	13 (35%)	23.7 ± 6.02

**Table 2 dentistry-13-00033-t002:** Clinical periodontal parameters (mean ± standard deviation) throughout time between the C and aPDT groups.

Periodontal Parameters	Groups	Baseline	3 Months	6 Months	12 Months
PD (mm)	aPDT	4.01 ± 0.96 ^aA^	2.57 ± 0.91 ^bA^	2.62 ± 0.89 ^bA^	2.54 ± 0.73 ^bA^
	C	4.16 ± 0.93 ^aA^	2.49 ± 0.61 ^bA^	2.59 ± 0.54 ^bA^	2.47 ± 0.75 ^bA^
CAL (mm)	aPDT	6.72 ± 0.65 ^aA^	4.58 ± 0.54 ^bA^	4.65 ± 0.55 ^bA^	4.59 ± 0.56 ^bA^
	C	6.84 ± 0.55 ^aA^	4.72 ± 0.44 ^bB^	4.74 ± 0.43 ^bA^	4.84 ± 0.51 ^bB^
BOP (% sites)	aPDT	57.35 ± 19.41 ^aA^	29.87 ± 11.66 ^bA^	36.82 ± 5.78 ^cA^	38.37 ± 15.74 ^cA^
	C	51.48 ± 19.17 ^aA^	34.63 ± 13.59 ^bA^	41.1 ± 15.27 ^cA^	36.09 ± 16.24 ^cA^
PI (% sites)	aPDT	85.67 ± 1.74 ^aA^	63.94 ± 3.11 ^bA^	74.33 ± 2.37 ^cA^	67.89 ± 6.32 ^cA^
	C	86.53 ± 1.64 ^aA^	65.37 ± 3.04 ^bA^	72.89 ± 2.96 ^cA^	69.74 ± 4.94 ^cA^

PD = probing depth; CAL = clinical level of insertion; BOP = bleeding on probing; PI = plaque index. Inter-group analysis—two-way ANOVA–Tukey/intra-group analysis—repeated measures ANOVA–Tukey/different lowercase letters = *p* < 0.05 for time/different uppercase letters = *p* < 0.05 for groups.

**Table 3 dentistry-13-00033-t003:** Number (mean ± standard deviation) of sites according to the probing depth range (mm).

Probing Depth Range	Group	Baseline	3 Months	6 Months	12 Months
1–3 mm	aPDT	136.83 ± 28.4 ^aA^	147.39 ± 21.5 ^bA^	147.39 ± 27.5 ^bA^	146.83 ± 21.4 ^bA^
	C	113.68 ± 45.1 ^aA^	121.68 ± 39 ^bB^	118.68 ± 36 ^bB^	121.5 ± 38.8 ^bB^
4–5 mm	aPDT	15.94 ± 10.6 ^aA^	6.33 ± 10.6 ^aA^	6.33 ± 10.6 ^aA^	7.39 ± 11.5 ^aA^
	C	20.26 ± 12.1 ^aA^	10.4 ± 10.5 ^aA^	12 ± 10.7 ^aA^	11.47 ± 11.4 ^aA^
6–7 mm	aPDT	1.78 ± 2.5 ^aA^	0.56 ± 1.3 ^aA^	0.61 ± 1.5 ^aA^	0.61 ± 1.5 ^aA^
	C	2.63 ± 3.2 ^aA^	0.95 ± 2 ^aA^	1.05 ± 2.3 ^aA^	1.05 ± 2.3 ^aA^
>7 mm	aPDT	0.17 ± 0.3 ^aA^	0 ± 0 ^aA^	0 ± 0 ^aA^	0 ± 0 ^aA^
	C	0.42 ± 0.6 ^aA^	0 ± 0 ^aA^	0 ± 0 ^aA^	0 ± 0 ^aA^

aPDT: antimicrobial photodynamic therapy group; C, control group. Inter-group analysis—two-way ANOVA–Tukey/Intra-group analysis—repeated measures ANOVA–Tukey/different lowercase letters = *p* < 0.05 for time/different uppercase letters = *p* < 0.05 for groups.

## Data Availability

Material will be available to interested researchers upon request.

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
