# Peer review of "Multiple Sessions of Antimicrobial Photodynamic Therapy Improve Periodontal Outcomes in Patients with Down Syndrome: A 12-Month Randomized Clinical Trial"

_dentistry, 2025, doi:10.3390/dj13010033_

Round 1
Reviewer 1 Report
Comments and Suggestions for Authors
Dear Authors, I have read your work on the use of a-PDT in patients with DS with interest. I have revised the manuscript in accordance with the provided outline.
Introduction This introduction provides a clear background on Down syndrome (DS) and its association with periodontal disease (PD), establishing the relevance of the study. The introduction lacks an explicit statement of the research question or hypothesis. While this implies that the study will investigate the effectiveness of aPDT in DS patients, it does not explicitly state the researchers' aims or objectives. - References 42-43, add a reference recent, for example, Pardo, A.; Butera, A.; Giordano, A.; Gallo, S.; Pascadopoli, M.; Scribante, A.Albanese, M. Photodynamic Therapy in Non-Surgical Treatment of Periodontitis: A Systematic Review and Meta-Analysis. Appl. Sci.2023,13, 1086. https://doi.org/10.3390/app13021086 - There is insufficient discussion on the research gap that this study aims to address, aside from a brief note on the paucity of long-term studies. Expanding on the limitations of the existing literature and how this study contributes to new knowledge would enhance its introduction. Methods: This section does not adequately elucidate the rationale behind certain exclusion criteria, which are crucial for understanding their potential impact on the study's outcomes and the selection of the study population. Results: - The manuscript lacks a comprehensive description or reference to the statistical methods employed for analysis within the text, mentioning only "Two-way ANOVA - Tukey / Intra-group analysis –Repeated measures ANOVA - Tukey" in the context of the tables. This would enhance the manuscript to briefly describe the selection of statistical tests and their appropriateness for this data. The presentation of the data in Tables 2 and 3 could be improved by including the actual measured values (e.g., mean PD and CAL at each time point) directly in the text for easier access and interpretation by the reader. There is no mention of the standard deviation or other measures of variability in the probing depth range in Table 3, which is essential for understanding the spread and reliability of the data. Discussion: The discussion commences directly by stating the prevalence and severity of periodontal disease in DS patients without providing a theoretical framework or hypothesis for the study. A brief introduction outlining the theoretical considerations or expectations based on prior research could establish a stronger foundation for the discussion. The manuscript notes that their study did not demonstrate a statistical correlation between decreased PD and CAL values, contrary to other studies, but did not thoroughly explore or explain this discrepancy. A more in-depth analysis or hypothesis regarding why their results differ could provide valuable insights and contribute to this paper's intellectual merit. - Suggestions for future studies are somewhat generic, focusing on larger sample sizes and the continuous use of antimicrobial solutions. Specific recommendations that stem directly from the findings or limitations of the present study would be more beneficial. Conclusion: It's acceptable
Author Response
Dear Authors, I have read your work on the use of a-PDT in patients with DS with interest. I have revised the manuscript in accordance with the provided outline.
R: Esteemed reviewer, we found the reflections presented to be enriching. We highlight the changes in the abstract marked in red.
Introduction This introduction provides a clear background on Down syndrome (DS) and its association with periodontal disease (PD), establishing the relevance of the study. The introduction lacks an explicit statement of the research question or hypothesis. While this implies that the study will investigate the effectiveness of aPDT in DS patients, it does not explicitly state the researchers' aims or objectives.
R: Esteemed reviewer, we highlight the information present in lines 76-79, which underwent a minor revision, as well as reference 62, which follows and complements these points.
- References 42-43, add a reference recent, for example, Pardo, A.; Butera, A.; Giordano, A.; Gallo, S.; Pascadopoli, M.; Scribante, A.Albanese, M. Photodynamic Therapy in Non-Surgical Treatment of Periodontitis: A Systematic Review and Meta-Analysis. Appl. Sci.2023,13, 1086. https://doi.org/10.3390/app13021086
R: We have included this reference, which has now become reference 61.
- There is insufficient discussion on the research gap that this study aims to address, aside from a brief note on the paucity of long-term studies. Expanding on the limitations of the existing literature and how this study contributes to new knowledge would enhance its introduction.
R: Esteemed reviewer, we would like to inform you that some items are briefly addressed in the introduction, specifically between lines 60 and 79. However, more detailed information can be found in the discussion.
Methods: This section does not adequately elucidate the rationale behind certain exclusion criteria, which are crucial for understanding their potential impact on the study's outcomes and the selection of the study population.
R: Esteemed reviewer, we have provided the reason for a clearer explanation of the exclusion criteria in line 98.
Results: - The manuscript lacks a comprehensive description or reference to the statistical methods employed for analysis within the text, mentioning only "Two-way ANOVA - Tukey / Intra-group analysis –Repeated measures ANOVA - Tukey" in the context of the tables. This would enhance the manuscript to briefly describe the selection of statistical tests and their appropriateness for this data.
R: Esteemed reviewer,
We appreciate your comments and suggestions. Regarding the observation about the description of the statistical methods, we agree that a more detailed explanation of the tests used could enhance the clarity of the manuscript. Although the use of 'two-way ANOVA - Tukey' and 'intra-group analysis - repeated measures ANOVA - Tukey' was briefly mentioned, we would like to clarify that all statistical procedures were conducted by an experienced statistician with specialized knowledge in the selection and application of the most appropriate methods.
The choice of two-way ANOVA with Tukey’s post-hoc test was based on the need to analyze the main effects and interactions between two independent variables, assessing differences between groups. This method is widely used and recognized for analyses where the variables are categorical and the data are parametric, allowing for the identification of significant differences between group means while controlling for Type I error.
On the other hand, the intra-group analysis - repeated measures ANOVA, also followed by Tukey’s test, was applied to assess the data evolution within each group over time, accounting for the dependency between observations from the same subject. The choice of this method was driven by the nature of the data, which consists of dependent samples collected at multiple time points, making repeated measures ANOVA the most appropriate statistical approach.
Overall, these analyses were chosen based on the characteristics of the data and the research questions, being suitable for the type of variables and the experimental design adopted. We emphasize that the tests used follow the standard recommendations in the literature for this type of analysis and were performed with rigor, aiming for the robustness and validity of the results.
We hope that this additional explanation helps clarify the choice of statistical methods and their suitability for the dataset, contributing to the improvement of the manuscript.
We once again thank you for your valuable suggestions.
The presentation of the data in Tables 2 and 3 could be improved by including the actual measured values (e.g., mean PD and CAL at each time point) directly in the text for easier access and interpretation by the reader. There is no mention of the standard deviation or other measures of variability in the probing depth range in Table 3, which is essential for understanding the spread and reliability of the data.
R: Esteemed reviewer, we have made the changes (as per lines 204 and 212).
Discussion: The discussion commences directly by stating the prevalence and severity of periodontal disease in DS patients without providing a theoretical framework or hypothesis for the study. A brief introduction outlining the theoretical considerations or expectations based on prior research could establish a stronger foundation for the discussion.
R: Esteemed reviewer, we appreciate the proposed suggestion. However, the structure of the initial part of the discussion was based on the alignment with the other references used in this study.
The manuscript notes that their study did not demonstrate a statistical correlation between decreased PD and CAL values, contrary to other studies, but did not thoroughly explore or explain this discrepancy. A more in-depth analysis or hypothesis regarding why their results differ could provide valuable insights and contribute to this paper's intellectual merit.
R: The explanation for this difference can be found between lines 201 and 205 with the inclusion of a brief text that enhances understanding.
- Suggestions for future studies are somewhat generic, focusing on larger sample sizes and the continuous use of antimicrobial solutions. Specific recommendations that stem directly from the findings or limitations of the present study would be more beneficial.
R: Esteemed reviewer, we respectfully disagree with this consideration. From line 316 to 345, the limitations (such as anatomical changes) were specifically addressed. Furthermore, the suggestions for future studies include, for example, the use of electric toothbrushes and even toothbrushes with three heads.
Conclusion: It's acceptable
R: We would like to express our sincere gratitude to the reviewer for the valuable comments and suggestions provided. We have carefully considered all the points raised, and the necessary revisions have been made in accordance with these recommendations. These modifications aim to enhance the clarity and quality of the manuscript, with the ultimate goal of ensuring its approval. We appreciate the constructive feedback, which has significantly contributed to the improvement of our work.
Reviewer 2 Report
Comments and Suggestions for Authors
This is a 12-month randomized clinical trial about periodontal outcomes in patients with Down syndrome. The study used PDT as a adjunct treatment with SRP, and achieved a significant effect on 3 mouths.
Author Response
Esteemed reviewer,
We would like to express our sincere gratitude to the reviewer for the valuable. We appreciate the constructive feedback, which has significantly contributed to the improvement of our work.
Reviewer 3 Report
Comments and Suggestions for Authors
The paper is interesting and the subject of the study has great clinical implications.
Nevertheless, there are some essential ideas to be reconsidered / clarified:
-in the abstract paragraph, the statement “Results: Both treatments promoted improvements in all clinical periodontal parameters (p > 0.05)” is correct, taking into consideration that p>0,05?
- In the materials and method chapter, line 114, it states “This is a double-blind study. Both patients and the statistician were blinded.” In this case, a double-blind study would have been if neither the participants nor the clinician who performed the periodontal examination knew who is receiving a particular treatment. The statistician should always be blinded; therefore, the explanation is not valid.
-optical fiber tip diameter of 0.000314 cm² is correct?
-description of the movement is incorrect, it cannot be applied to the proximal pockets: “sweeping mode irradiation (medial-distal movements) into the periodontal sulcus or pocket (40 s for buccal 140 and 40 s for lingual sites)”;
-Figure 1-“clinical meansurements” spelling error
-there is a lack of coherence between the explanations in the text and those in the figure, as follows: in the text it is written “in the aPDT group, aPDT was performed on all teeth and repeated after 3, 7, 146 and, 14 days”, while in the figure, on days 3, 7 and 14, SRP or SRP+aPDT appears to be performed. The protocol is unclear. If SRP was really performed every 2-4-7 days, what is the scientific support for this type of protocol?
-at the same time, it is specified that “Patients in both groups received supragingival dental prophylaxis every three weeks throughout the study period.” and “Patients were enrolled in a hygiene program tailored to their individual needs and received reinforcement of these instructions at each appointment.” Does this mean that the working protocol was not uniformly applied to all patients?
-on the other hand, an important source of errors could be the following variation: “suboptimal treatment of these areas, as the extent of optical fiber penetration and photosensitizer irrigation was dependent on the patient’s behavioral response.”
-this affirmation is based on the present study or it corresponds to the cited source? “At all evaluation periods, there were high values for probing depth (PD), clinical attachment level (CAL), plaque index (PI), and bleeding on probing (BOP), complicating the classification of the clinical picture of periodontal health [14]”.
-the same question applies to the affirmation “The application of aPDT to the full mouth via optical fibers inserted into the periodontal sulcus or pockets caused mild discomfort, which required multiple appointments to complete the treatment at all sites/teeth. Despite this, patients with DS often showed good acceptance of this therapy [26].” In addition, the decontamination in different stages of the oral cavity can represent a source of errors, since recontamination is very possible, between sessions.
-overall, the presented results seem to be inconsistent and with many factors that influenced them and that were not taken into account at the beginning of the study, in an attempt to reduce them. Also, the presentation of these results can be greatly improved, for clarity.
-the conclusions are not in agreement with the discussions, especially regarding the long-term effects
Author Response
The paper is interesting and the subject of the study has great clinical implications.
Nevertheless, there are some essential ideas to be reconsidered / clarified:
R: Esteemed reviewer, we found the reflections presented to be enriching. We highlight the changes in the abstract marked in red.
-in the abstract paragraph, the statement “Results: Both treatments promoted improvements in all clinical periodontal parameters (p > 0.05)” is correct, taking into consideration that p>0,05?
R: Esteemed reviewer, thank you for your consideration. The symbol was indeed incorrect. We have made the correction in the abstract and replaced it with the correct symbol (p < 0.05).
In the materials and method chapter, line 114, it states “This is a double-blind study. Both patients and the statistician were blinded.” In this case, a double-blind study would have been if neither the participants nor the clinician who performed the periodontal examination knew who is receiving a particular treatment. The statistician should always be blinded; therefore, the explanation is not valid.
R: We refer to the term 'double-blind' when both the participants and the statistician (or data analyst) are unaware of the group assignments of the participants. This blinding is implemented to minimize bias from both the participants and the researchers or statisticians conducting the data analysis.
The definition of a double-blind study can be justified based on scientific methodology books that address practices in clinical trials and randomized controlled studies. One example is the book Fundamentals of Clinical Trials by Friedman, Furberg, and DeMets (2010), which discusses the concept of blinding in clinical studies.
"In a double-blind study, neither the participants nor the investigators (or those analyzing the data) know which treatment the participants are receiving. This is done to prevent bias in the treatment administration, data collection, and analysis."
This excerpt clarifies that in a double-blind study, both the participants and the investigators (or data analysts) are blinded to the treatment allocation, with the aim of minimizing any bias in treatment administration, data collection, and result analysis. Therefore, it is entirely appropriate to refer to a study as double-blind when neither the participants nor the statistician knows the group assignments. This enhances the objectivity in data analysis and helps ensure that the results are not influenced by expectations or biases from either the participants or the analysts. This reference has been included in our study [63].
-optical fiber tip diameter of 0.000314 cm² is correct?
R: Yes, it is correct.
-description of the movement is incorrect, it cannot be applied to the proximal pockets: “sweeping mode irradiation (medial-distal movements) into the periodontal sulcus or pocket (40 s for buccal 140 and 40 s for lingual sites)”;
R: Esteemed reviewer, indeed, the information was incomplete. Therefore, we made the necessary adjustments, and the information has been added to the manuscript (line 141).
-Figure 1-“clinical meansurements” spelling error
R: We apologize for the error and have made the correction.
-there is a lack of coherence between the explanations in the text and those in the figure, as follows: in the text it is written “in the aPDT group, aPDT was performed on all teeth and repeated after 3, 7, 146 and, 14 days”, while in the figure, on days 3, 7 and 14, SRP or SRP+aPDT appears to be performed. The protocol is unclear. If SRP was really performed every 2-4-7 days, what is the scientific support for this type of protocol?
R: As explained in the article (line 146), all patients had their needs addressed, and we considered this as 'day zero'. Subsequently, the therapy applications were performed according to each group.
The protocol is scientifically supported and based on the methodology of Lulic et al., 2009.
Reference 51: Lulic, M., Leiggener, G., Salvi, G. E., Ramseier, C. A., Mattheos, N., & Lang, N. P. (2009). One-year outcomes of repeated adjunctive photodynamic therapy during periodontal maintenance: a proof-of-principle randomized-controlled clinical trial. Journal of Clinical Periodontology, 36, 661-666.
We emphasize that prior to the application of antimicrobial photodynamic therapy, it is crucial to remove/disorganize the dental biofilm present (which is why SRP is necessary). To match the actions between the groups, the control group received only SRP.
-at the same time, it is specified that “Patients in both groups received supragingival dental prophylaxis every three weeks throughout the study period.” and “Patients were enrolled in a hygiene program tailored to their individual needs and received reinforcement of these instructions at each appointment.” Does this mean that the working protocol was not uniformly applied to all patients?
R: The hygiene instructions were applied in a standardized manner to all patients. However, the individual needs of each patient were noted and evaluated during each follow-up. For example, some patients had more biofilm in the interproximal areas, and in these cases, the use of dental floss was emphasized. These details can be found from line 169 to line 174.
-on the other hand, an important source of errors could be the following variation: “suboptimal treatment of these areas, as the extent of optical fiber penetration and photosensitizer irrigation was dependent on the patient’s behavioral response.”
R: We consider this issue as an aspect of dental care for patients with Down syndrome. We observed that the introduction of the optical fiber into the periodontal pocket causes a slight discomfort, which, depending on the patient's cognitive threshold, could compromise the success of the procedure. For example, some patients may not allow for reapplication or completion of antimicrobial photodynamic therapy. We have raised this point so that future studies can explore alternative methods for dye activation that cause less discomfort to the patient.
-this affirmation is based on the present study or it corresponds to the cited source? “At all evaluation periods, there were high values for probing depth (PD), clinical attachment level (CAL), plaque index (PI), and bleeding on probing (BOP), complicating the classification of the clinical picture of periodontal health [14]”.
-the same question applies to the affirmation “The application of aPDT to the full mouth via optical fibers inserted into the periodontal sulcus or pockets caused mild discomfort, which required multiple appointments to complete the treatment at all sites/teeth. Despite this, patients with DS often showed good acceptance of this therapy [26].”
R: We will address these two statements as they present the same outcome. These data pertain to our article and are consistent with the references listed below, following an indirect citation.
In addition, the decontamination in different stages of the oral cavity can represent a source of errors, since recontamination is very possible, between sessions.
R:This is an important observation that we have brought forward to be addressed in our discussion. We discuss and suggest this issue in lines 340-342, as well as in lines 273-280.
-overall, the presented results seem to be inconsistent and with many factors that influenced them and that were not taken into account at the beginning of the study, in an attempt to reduce them. Also, the presentation of these results can be greatly improved, for clarity.
R: The results presented follow a pattern consistent with other studies, as cited in our reference list, and are also supported by the systematic reviews mentioned. We acknowledge that there are several limitations inherent in clinical studies, and we made every effort to minimize them. One of the main focuses was the control of dental biofilm (which has already been mentioned in a previous section). Therefore, considering that this is dental care for a patient with Down syndrome, biases were extensively minimized, and the remaining ones are inherent and also present in other clinical studies in the literature.
-the conclusions are not in agreement with the discussions, especially regarding the long-term effects
R:Esteemed reviewer, we respectfully disagree with this observation. Our article presents a clear objective, which was promptly addressed in the conclusion.
We would like to express our sincere gratitude to the reviewer for the valuable comments and suggestions provided. We have carefully considered all the points raised, and the necessary revisions have been made in accordance with these recommendations. These modifications aim to enhance the clarity and quality of the manuscript, with the ultimate goal of ensuring its approval. We appreciate the constructive feedback, which has significantly contributed to the improvement of our study.
Reviewer 4 Report
Comments and Suggestions for Authors.

Author Response

(The authors gave the same response as above.)

Round 2
Reviewer 3 Report
Comments and Suggestions for Authors
Dear authors, with respect, I emphasize the following:
- although you responded punctually to each observation, confirming the limitations of the study with articles of the same type, you did not manage to improve your manuscript almost at all (12 serious observations materialized, in the text, in 3 extremely minor corrections; for instance, the discussion part was not altered at all).
-What is the contribution of your study, compared to the cited studies, which have already raised serious problems regarding the limitations of this type of study?
-optical fiber tip diameter of 0.000314 cm² is correct? I asked this question in the previous review, and your answer was it is correct. Once again, I draw your attention to the fact that it is impossible. Cm2 denotes an area, not a diameter. Therefore, the correct information regarding the diameter must be inserted, which cannot be the displayed value.
- Regarding the double blinded study, I quote from your response: "In a double-blind study, neither the participants nor the investigators (or those analyzing the data) know which treatment the participants are receiving. This is done to prevent bias in the treatment administration, data collection, and analysis. This excerpt clarifies that in a double-blind study, both the participants and the investigators (or data analysts) are blinded to the treatment allocation, with the aim of minimizing any bias in treatment administration, data collection, and result analysis." This is the definition, which has to be individualized depending on the study. In your case, the elimination of the risk of bias is based on an objective measurement of the periodontal indices (the statistician is always blinded). Therefore, the examiner had to be blinded!
-once again I return to the observations regarding the discussion chapter, which requires restructuring, for a better understanding of the information (observations 6-10 from the previous review, which were aimed at clarifying the presentation of the discussions)
Author Response
Esteemed reviewer, we found the reflections presented to be enriching. We highlight the changes in the abstract marked in blue. In red, we have kept the changes initially indicated in the initial evaluation.
- although you responded punctually to each observation, confirming the limitations of the study with articles of the same type, you did not manage to improve your manuscript almost at all (12 serious observations materialized, in the text, in 3 extremely minor corrections; for instance, the discussion part was not altered at all).
R: Dear reviewer, we have made significant changes throughout the text, particularly in the discussion section.
-What is the contribution of your study, compared to the cited studies, which have already raised serious problems regarding the limitations of this type of study?
R: Regarding the contributions of our study, we highlight the long-term evaluation (up to 12 months), the application of multiple aPDT sessions, and the results (which demonstrated improvements in CAL at 3 and 12 months) along with statistically significant changes in the number of sites with shallow pockets.
These aspects were further emphasized through revisions in the discussion section, as well as in the restructuring of the conclusion.
-optical fiber tip diameter of 0.000314 cm² is correct? I asked this question in the previous review, and your answer was it is correct. Once again, I draw your attention to the fact that it is impossible. Cm2 denotes an area, not a diameter. Therefore, the correct information regarding the diameter must be inserted, which cannot be the displayed value.
R: It was indeed incorrect. We apologize for the reiteration of the error. This issue has been addressed and is now included in the article (line 141).
- Regarding the double blinded study, I quote from your response: "In a double-blind study, neither the participants nor the investigators (or those analyzing the data) know which treatment the participants are receiving. This is done to prevent bias in the treatment administration, data collection, and analysis. This excerpt clarifies that in a double-blind study, both the participants and the investigators (or data analysts) are blinded to the treatment allocation, with the aim of minimizing any bias in treatment administration, data collection, and result analysis." This is the definition, which has to be individualized depending on the study. In your case, the elimination of the risk of bias is based on an objective measurement of the periodontal indices (the statistician is always blinded). Therefore, the examiner had to be blinded!
R: We appreciate your comment regarding the study procedure and would like to clarify and reinforce that, in our study design, the protocol was strictly followed to ensure the elimination of any bias. Specifically, the study was conducted in a double-blind manner, meaning that both the participants and the statistician (responsible for data analysis) were unaware of the treatment or procedures applied.
The aim of this approach was to ensure that statistical analyses and data collection were not influenced by expectations or biases related to the administered treatments. The blinding of the statistician is a crucial element to guarantee that the results obtained are exclusively based on the collected variables and data, without any subjective interference.
We understand that, in a double-blind study, the blinding of the examiners is also a recommended practice. However, in our case, the primary measure for bias control was the blinding of the statistician, and the objective and measurable data collection, such as periodontal indices. Ensuring that the statistician was unaware of the treatments applied was critical to maintaining the integrity of the data analysis and results. We utilized a reference that has been included in the text [63].
Additionally, when participants, examiners, and statisticians are unaware of the procedures carried out, the study is classified as triple-blind, which provides an even more stringent control over potential biases. The scientific literature supports that, in this model, the results are even more reliable, as it eliminates any possibility of bias both in treatment administration and in data collection and statistical analysis.
We have added a paragraph in the discussion addressing this (starting from line 355-367).
We hope this explanation helps clarify our methodological approach, and we remain available to provide further details if necessary.
-once again I return to the observations regarding the discussion chapter, which requires restructuring, for a better understanding of the information (observations 6-10 from the previous review, which were aimed at clarifying the presentation of the discussions)
R: We would like to emphasize that we have made numerous changes to the text, which are reflected in the revised version, addressing the points raised.
These modifications aim to enhance the clarity and quality of the manuscript, with the ultimate goal of ensuring its approval. We appreciate the constructive feedback, which has significantly contributed to the improvement of our work.
Round 3
Reviewer 3 Report
Comments and Suggestions for Authors
The observations made were addressed.